# Theranostics Nanomedicine Applications for Colorectal Cancer and Metastasis: Recent Advances

**DOI:** 10.3390/ijms24097922

**Published:** 2023-04-27

**Authors:** Phanindra Babu Kasi, Venkata Ramana Mallela, Filip Ambrozkiewicz, Andriy Trailin, Václav Liška, Kari Hemminki

**Affiliations:** 1Laboratory of Translational Cancer Genomics, Biomedical Center, Faculty of Medicine in Pilsen, Charles University, Alej Svobody 1665/76, 323 00 Pilsen, Czech Republic; 2Laboratory of Cancer Treatment and Tissue Regeneration, Biomedical Center, Faculty of Medicine in Pilsen, Charles University, Alej Svobody 1665/76, 323 00 Pilsen, Czech Republic; 3Department of Surgery, University Hospital in Pilsen and Faculty of Medicine in Pilsen, Charles University, Alej Svobody 80, 323 00 Pilsen, Czech Republic; 4Department of Cancer Epidemiology, German Cancer Research Center, Im Neuenheimer Feld 280, 69120 Heidelberg, Germany

**Keywords:** colorectal cancer, nanomedicine, theranostics, clinical status, cancer therapy

## Abstract

Colorectal cancer (CRC) is the third most common cancer worldwide, and metastatic CRC is a fatal disease. The CRC-affected tissues show several molecular markers that could be used as a fresh strategy to create newer methods of treating the condition. The liver and the peritoneum are where metastasis occurs most frequently. Once the tumor has metastasized to the liver, peritoneal carcinomatosis is frequently regarded as the disease’s final stage. However, nearly 50% of CRC patients with peritoneal carcinomatosis do not have liver metastases. New diagnostic and therapeutic approaches must be developed due to the disease’s poor response to present treatment choices in advanced stages and the necessity of an accurate diagnosis in the early stages. Many unique and amazing nanomaterials with promise for both diagnosis and treatment may be found in nanotechnology. Numerous nanomaterials and nanoformulations, including carbon nanotubes, dendrimers, liposomes, silica nanoparticles, gold nanoparticles, metal-organic frameworks, core-shell polymeric nano-formulations, and nano-emulsion systems, among others, can be used for targeted anticancer drug delivery and diagnostic purposes in CRC. Theranostic approaches combined with nanomedicine have been proposed as a revolutionary approach to improve CRC detection and treatment. This review highlights recent studies, potential, and challenges for the development of nanoplatforms for the detection and treatment of CRC.

## 1. Introduction

Colorectal cancer (CRC) is the third most common cancer in the world and the second leading cause of death in 2020, with 1.93 million new cases and 930,000 deaths [1]. In 2020, nearly 520,000 new cases and 245,000 deaths in Europe were registered [2]. The estimated number of new cases in 2040 is 3.2 million, representing a 63% increase over the estimated number of cases in 2020. CRC was the most malignant in 18 of 185 countries, and it ranked second or third in another 78 countries [1]. Obesity, crimson meat, smoking, and excessive drinking are all generic CRC risk factors. On the other hand, lifestyle changes, diet, and genetic mutations play a role in the development of the disease. However, dietary fiber and aspirin are considered protective factors [3,4,5,6].

CRC treatment is determined by the disease’s stage [7]. Surgical resection and postoperative adjuvant chemotherapy are the most effective treatments in the early stages of CRC. These treatments can successfully remove any remaining micrometastases, remove any local implants during surgery, and reduced the probability of recurrence [3]. Blood stool tests, X-ray, colonoscopy, CT colonography, magnetic resonance imaging (MRI) scans, stool DNA tests, flexible sigmoidoscopy, and barium enema have all been developed for CRC diagnosis. However, CRC diagnosis is challenging due to the large number of gaps that exist in addition to over-testing, over diagnosis, overtreatment, non-specificity, and the heterogeneous nature of CRC [8]. The number of possible therapeutic targets is constantly growing as a result of numerous factors that influence and contribute to the onset and spread of the disease. In the end, this targeting potential will pave the way for the development of a successful method for the management of disease and better patient care. For several decades, research has focused on developing new approaches and techniques for studying cancer, particularly in the areas of detection and early cancer therapy to reduce mortality.

Nanoparticles have the potential to be employed for both diagnostic and treatment at the same time. Researchers from all over the world are exploring the use of nanoparticles because of these fascinating characteristics. The goal of this narrative review is to highlight current knowledge as well as the most recent discoveries and clinical and research findings concerning the use of nanoparticles (NPs) in CRC treatment. We focused on the theranostics applications of various types of nanomaterials, including liposomal nanoparticles, polymeric nanoparticles, bio-nanoparticles, and metal nanoparticles that have shown significant potential for applications in CRC detection and therapy (Figure 1).

## 2. Colorectal Liver Metastasis

CRC most commonly metastasizes to the liver (CRLM); historically, the surgical management approach to CRLM has evolved significantly [9]. Liver metastasis may be located in the portal vein, which connects the colorectal and liver and is associated with abundant blood supply [10,11]. Standard therapeutic protocols for CRLM are curative resection, embolization, and chemotherapy, However, due to the location and size of the tumor, the presence of extrahepatic disease, the patient’s comorbidities, or unresectable disease, surgery is only possible in 10–20% of cases, and less than 5 year survival rate is about 30% [11,12]. It has shown to be challenging to target metastases effectively. Targeting malignant cells within locations of metastasis is challenging due to biological hurdles that must be overcome. The mechanisms driving metastasis are still being studied, which is allowing for the development of new treatments. Metastasizing malignancies can now be treated in ways that were previously unachievable thanks to targeted drug delivery using technologies made possible by nanotechnology. Immunotherapy has the potential to treat colon cancer since it triggers the body’s immunological system to attack malignant tumors.. Immune checkpoint inhibitors (ICIs) have swiftly become a popular treatment choice for a variety of solid tumors due to their higher efficacy. By focusing on receptor or ligand checkpoint proteins to inhibit immunosuppressive tumor signaling, such as programmed cell death 1 (PD-1), PD-1 ligand 1 (PDL1), and cytotoxic T lymphocyte antigen 4 (CTLA-4), ICIs restore the anticancer immune response [13]. Zhao et al. summarized several check point inhibitors in CRC [13]. The progress and recent advances in cancer immunotherapy in CRC are summarized in Table 1.

## 3. Applications of Nanotechnology for CRC Diagnosis and Treatment

Several categories of molecules, such as toxins, nucleic acids, radioisotopes, and hydrophobic drugs, are being used in cancer treatment. However, because of their extensive off-target effects and instability in the biological environment, they are not suitable for systematic injection into patients. The use of nanomaterials (NMs) as medication delivery systems was originally described in 1986, and it has since been established that NMs passively accumulate within tumoral tissues. NMs are regarded as therapeutic agents for cancer because of their high specificity and longer blood circulation duration. There are several families of inorganic and organic NPs with a wide range of sizes, structures, and chemical compositions now accessible [22,23]. Inorganic nanoparticles, such as carbon nanotubes, gold nanoparticles, iron oxide Nps, nonoshella and quantum dots, have good stability and minimal biodegradability for diagnostic usage, making them suitable for cancer imaging. Organic nanoparticles, such as liposomes, dendrimers, nano emulsions, micelle and ferrin, have a lesser stability but higher biocompatibility, and there are many ways to add functional groups for drug delivery (Figure 1).

## 4. Iron Oxide Nanoparticles (IONS)

Iron oxide nanoparticles (IONS) or superparamagnetic iron oxide nanoparticles (SPIONS) have attracted the attention of researchers due to their wide range of applications in the biomedical field and also exhibit biocompatibility and non-toxicity [24,25]. The magnetic core to the iron oxide nanocrystals ranges from 1 to 100 nm in diameter. IONS have a variety of biological characteristics because of their small dimensions, slow rate of deposition, effective surface area, and simplicity of cellular transport. The IONS loaded with 5-fluorouracil (5-FU) with magnetic hyperthermia efficiently reduces the growth of heterotopic human colon tumor growth in mice models [26]. The tertiary complex of Epirubicin-5TR1 aptamer and SPION enables tumor detection by MRI and efficient delivery of Epirubicin to the murine C26 murine colon carcinoma cells [25]. Human colorectal cell lines (HCT 116) exposed to IONS coated with chitosan induced increased reactive oxygen species (ROS), which activates Caspase 9/3 to cause apoptotic cell death [27].

## 5. Quantum Dots

Quantum dots are zero-dimensional nanomaterials with particle sizes ranging from 2–10 nm. The fluorescence emission properties of quantum dots are size-dependent. The unique optical and chemical properties of quantum dots become noble materials for cancer research, particularly CRC. The extended excitation wavelengths, photostability, narrow emission band, and high quantum efficiency are advantageous properties of quantum dots and have made them useful in biomedical applications [28,29]. These quantum dots are used as fluorescent markers for various in vitro and in vivo cancer studies, but in vivo studies are limited to non-targeted or xenograft labeling [30,31]. QDot655 targeted to vascular endothelial growth factor receptor 2 (VEGFR2) showed its capability to detect VEGFR2 expression in vivo CRC tumors [32]. Two cell lines, MC38 murine colon cancer, and RAW 264.7 mouse macrophages cell line, were used to study the interaction between IgG-functionalized Boron carbide (B4C) NPs, synthesized from B4C powder produced by the direct reaction between boron and soot under argon flow. Analysis using flow cytometry revealed that macrophages absorbed more fluorescently tagged NPs than cancer cells [33,34].

## 6. Poly (Lactic-co-glycolic Acid) NPs/Nano Cells

The US FDA has cleared the use of poly (lactic-co-glycolic acid), or PLGA, a natural and synthetic polymer NP, for medication delivery by oral administration. PLGA NPs are very stable in biological fluids, physically and chemically stable, and protect encapsulated medicines from enzymatic destruction. PLGA NPs can also enclose macromolecules, which increases the molecules’ thermal stability and encourages prolonged release [35,36]. PLGA is frequently used as a drug carrier because of its biodegradable and biocompatible qualities, as well as its capacity to encapsulate hydrophilic and hydrophobic medicines [37,38].

Docetaxel loaded with PEGylated-PLGA nanocapsules and SPIONs successfully restricted tumor growth in CT26 colon cancer [39]. The proliferation of the colon cancer cell line HT-29 was reduced by 5-FU encapsulated in PLGA-NPs by incresing the intracellular drug concentration in cancer cells [40]. 5FU-perfluorocarbons loaded into EGF-functionalized PLGA NPs efficiently inhibited colon tumor growth [41]. NP-PEG-FA/17-AAG (17 AAG incorporated into PLGA/PLA-PEG-FA-NPs synthesized through one-step surface functionalized techniques) has improved the oral bioavailability and efficiently treated ulcerative colitis and its associated tumors [42]. PHBV/PLGA NPs loaded with 5-FU are a promising nano-drug delivery system in colon cancer treatment [43].

## 7. Dendrimers

Dendrimers are 3D chemical structures and highly branched globular molecules that are very useful for nanopharmaceuticals because their backbones are biodegradable. Dendrimers’ adaptability in the administration of anticancer drugs and their theranostic uses in cancer therapy are widely established. Drugs can be delivered intracellularly via anticancer-binding dendrimers that bypass efflux transporters, increasing the bioavailability of charged molecular payloads. Additionally, these dendrimers are employed in the delivery of diagnostic chemicals for tumor-specific imaging [44,45]. The use of chemotherapy to treat CRC has not been successful because just a small proportion of the drugs are effective at reaching their target areas in tumors [46].

Cancer cells called circulating tumor cells (CTCs) move around and are primarily responsible for the spread of tumors. Considering the importance of CTCs as indications of poor outcomes, several approaches, including microfluidic-based and size-based filtering, etc., have been employed unsuccessfully to extract CTCs from sizable populations of interfering cells. Dendrimers attached to antibodies have been utilized in many techniques to identify colorectal CTCs, such as the capture of colon cancer HT29 cells using sialyl Lewis X antibodies and PAMAM dendrimers [47]. In addition to diagnostic applications, dendrimers have also been reported to be used for in vitro anticancer experimental therapy. G4-PAMAM dendrimers conjugated to capecitabine have been reported to reduce tumors and reduce the by-product of capecitabine [48]. In colorectal C26 and HT29 cells, Au NPs of PAMAM dendrimers coupled with curcumin showed significant cell uptake, internalization, and cytotoxicity [49]. The PEGylated PAMAM dendrimers were loaded with camptothecin and AS1411 (anti-nucleolin aptamer) for site-specific targeting of CRC cells [50]. The active irinotecan metabolite, SN-38, is coupled with L-lysine dendrimers that include polyoxazolines to increase effectiveness and reduce adverse effects [51]. Oxaliplatin-containing PAMAM-G4 dendrimers exhibit improved targeting effects on CRC cells that express the folic acid receptor in vitro. Gemcitabine-loaded YIGSR-CMCht/PAMAM dendrimer NPs caused target death in HCT-116 cancer cells [52,53].

## 8. Carbon Nanotubes

The carbon nanostructures known as carbon nanotubes (CNTs) play a variety of roles as carriers in the administration of drugs, gene therapy, immunotherapy, and diagnostics. They possess excellent optical qualities, thermal conductivity, chemical stability, and functionality. CNTs are hexagonal nanostructures made of tiny tubular carbon atoms that have special physicochemical features [53]. Based on the number of carbon atoms in the sheet, CNTs are classified as single-walled CNTs or multi-walled CNTs. Several studies reported the effectiveness of CNTs in cancer diagnosis and therapy, and CpG-conjugated CNTs to increase CpG uptake in mouse colon cancer cells and activate nuclear transcription factor kappa β signaling. The conjugated CpG-CNT successfully attenuated the local xenograft tumor growth and liver metastasis [54]. In contrast to their non-functional counterparts, single-walled CNT/II-NCC fluorescein-functionalized composites displayed intrinsic activity on cancer cell lines (Colon cancer cell lines-Caco-2). Through receptor-mediated endocytosis, the single-walled CNT-binding antibody C225 binds to CRC cells that express the epidermal growth factor receptor. Single-walled CNT nano biocomposites improved colon cancer cell killing during photodynamic treatment [55,56,57].

## 9. Liposomes

Liposomes are synthetic, lipid-based vesicle carriers that are non-toxic. In 1961, the FDA approved the use of liposomes as nanocarriers for the delivery of drugs. They have a tiny, spherical aqueous core and a phospholipid bilayer structure [58]. The main advantage and characteristic of NPs is their compact size, which supports the application of particles for the efficient and targeted administration of drugs for the treatment and diagnosis of diseases [59]. These particles also have a few modest adverse effects. Liposomes are primarily employed as delivery vehicles for nucleic acids, proteins, and peptides among other NPs. There are three different types of liposomes depending on how they function: thermo-sensitive/pH-sensitive/magnetic liposomes, active targeting, and long-circulating or stealthy liposomes [60]. For instance, Thermodox^®^ is a thermosensitive liposome used to treat CRC, while doxorubicin (Doxil), DaunoXome, and Marqibo^®^ are FDA-approved liposomal medications [61]. Over the last ten years, several investigations for the encapsulation of boron compounds into liposomes have been conducted. Low-invasive cancer treatment with boron neutron capture is based on the neutron fission process that takes place when molecules containing ^10^B are exposed to thermal neutrons. PEGylated liposomes functionalized with transferrin (TF-PEG liposomes) with sodium borocaptate (BSH) with concentrations of 107–123 nm/26–30 µg/µmol of lipid 6–8% were verified for the specificity of the TF-receptor mediated binding in vitro in colon carcinoma cells. In vivo colon-26 mouse colon carcinoma cells and ex vivo colon-26-bearing male BALB/c mice exhibited decreased tumor growth rate [33,62].

## 10. Gold NPs

The most stable noble NPs are considered to be gold NPs. They may be nanostructured into shapes such as pyramids, bowls, cubes, spheres, rods, flowers, branches, wires, and cages. Gold NPs are safer, more effective in delivering drugs, and more focused on targeting cancer, thanks to precise surface coating [63]. Gold NPs improved the efficacy of cisplatin delivery and effectively decompressed CRC vessels [64]. Similar to nanoemulsions, gold NPs can bind multiple molecules such as antibodies, nucleic acids, proteins, enzymes, and fluorescent dyes. These factors enhance the properties of gold NPs, such as stability, biocompatibility, and functionalization in the medical field [65].

## 11. Nanoemulsions

As the name suggests, nanoemulsion systems are composed of oil, water, and surfactants to create a clear colloidal solution. They are characterized by low toxicity, high stability, good heat sensitivity, pH sensitivity, and good potency. Today, CRC cells are the target of anti-angiogenic medications. These substances are vulnerable to barriers and produce toxicity and resistance. Nanoemulsions can easily penetrate through angiogenic tissue in tumor niches to kill cancer cells. To distribute drugs that are not soluble in water and have a hydrophobic core, nanoemulsion devices are employed. Nanoemulsions have been coupled with different antibodies for selective and precise targeting, efficient drug delivery, and therapeutic efficacy [63]. Nanoemulsions bind polyethylene glycol as well as antibodies. DNA complexes and polyelectrolyte complex microcells both have more promise as therapies for cancer. An emulsifier that is frequently used and approved is Tween-80 [66,67].

## 12. Other NPs

Therapeutic drugs used in conventional cancer therapies harm the immune system and have several side effects. By encapsulating therapeutic chemicals and then delivering tailored medications to tumor niches, nano-drug delivery systems minimize adverse effects. These systems, such as Plitidepsin-unloaded polymersomes, near-infrared fluorescent proteinoid-poly (L-lactic acid), P (EFG-proteinoid-poly (L-lactic acid)] random copolymer, proteinoid-proteinoid-poly (L-lactic acid) copolymer, etc., not only lessen toxicity in the body but are also highly stable, biocompatible, and effective. Several copolymers are efficient against CRC cells and have no negative side effects. For the treatment of CRC, curcumin has been nano-formulated as micelles, nanogels, liposomes, NPs, and cyclodextrins [68].

## 13. Targeted NPs in CRC Research

NPs are being looked at as a way to improve the effectiveness of chemo-treatments by specifically targeting tumor cells. This means that fewer side effects are likely to occur for patients, and the drugs may be more effective in killing the tumor. Nano-drug delivery systems are better than free drugs because they have increased bioavailability, better tissue targeting, and fewer side effects [69]. They are being used to address a variety of illnesses, including eye conditions [70], inflammation of the colon [71], osteoporosis [72], Alzheimer’s disease [73], and ischemic stroke [74]. Chemotherapy medications can be used to treat cancer both primary tumors and metastases. However, these drugs have many limitations. For example, they often have little selectivity, meaning they cannot easily reach tumor cells. Additionally, high doses of these drugs can sometimes cause side effects.

Targeted nanoparticles are made up of either natural or synthetic materials. Natural nanoparticles, such as those made from biological materials such as cells or vesicles, have more biological components and are more permeable to medication. Synthetic nanoparticles, such as those made from inorganic materials, are more permeable and have better drug-delivery properties than other types of carriers. Nanocarriers need to be compatible with cells and biodegradable so they do not have a big impact on their growth and metabolism [75]. They can be loaded with lots of active substances, such as chemotherapeutics drugs to treat cancer, RNA molecules that can silence genes, proteins, and contrast agents. This technology is advancing quickly, and we are expecting nanocarriers to improve the effectiveness of cancer treatments [76]. By altering the pharmacokinetics and tissue distribution of chemotherapeutic medications, preventing tumor growth, and lowering drug toxicity to healthy tissues, targeted NPs as DDSs can assist in increasing the specificity of these therapies [77]. NPs modified with a target can help to improve the concentration of drugs in tumor tissues, which can help to treat tumors and improve tumor diagnosis and treatment. NP therapy is good for delivering drugs to tumors specifically, and it prevents the release of the drugs into the general population. This can improve the pharmacokinetics and pharmacodynamics of the drugs and help overcome tumor cell resistance mechanisms [78].

NPs made of special materials can protect the contents from being damaged by outside forces and can provide controlled, sustained release of the drug. Natural polymers such as chitosan, gelatin, and alginate can be used to create the NPs, while synthetic polymers such as polyethylene glycol (PEG), polylactic acid (PLA), and polycaprolactone (PCL) can also be used. While natural polymers are more environmentally friendly, synthetic polymers are easier to create and modify and have better biodegradability [79].

Passive targeting helps increase the effectiveness of the Enhanced Permeability and Retention (EPR) effect by exploiting the basement membrane in tumors with an incomplete vascular system. This can also release the drug at specific places for targeted drug delivery by making use of the unique pH, enzyme environment, and intracellular reducing environment of the tumor site. By altering the surface of NP carriers, effective targeting is made possible. Antibodies, peptides, sugar chains, and nucleic acid aptamers are just a few examples of probe molecules that can precisely attach to the target molecule and are connected to the carrier surface by chemical or physical means. Utilizing ligands that bind to tumor cells can boost the percentage of nano-drugs that reach the target [80].

Designing targeted nanoparticles for cancer treatment requires considering factors such as biocompatibility and degradability, stability of physical and chemical properties, and the aggregation of nanoparticles at the tumor site. Based on these considerations, a variety of heavily designed targeted nanoparticles have been studied extensively in cancer chemotherapy.

## 14. Passive Targeted NPs

### 14.1. Facile NPs

Through enhanced EPR effects, facile NPs passively deliver chemotherapy to malignant tissues. Normal tissues have a tightly packed microvascular endothelial barrier that prevents large molecules and lipid particles from easily crossing the vascular wall. In contrast, tumor tissues have a wide vascular wall gap and poor structural integrity, which makes them more susceptible to exposure to nano-drugs with a small diameter [81].

Polymers such as PLGA can be found in nature, and they are made up of many different kinds of molecules. Albumins and chitosans can be produced using polymer materials such as PLGA. Some types of polymer substances are made to be biodegradable, and they have good compatibility with other molecules. PLGA-NPs are nanoparticles made from PLGA that are designed to be taken up by cancer cells and release chemotherapeutic drugs and other substances into the cancer cells. The PLGA-NPs were coated with cholesterol to make them more likely to be absorbed by tumors. In comparison to free oxaliplatin, the oxaliplatin-loaded NPs exhibited greater pro-cancer cell apoptosis and protection against non-tumor cells [82]. In addition, microfluidics allows for more precise control of drug release, which can reduce adverse effects due to drug dose [83]. The negative charge on the surface of polylactic-glycolic acid (PLGA) nanoparticles negatively affects their rate of cellular uptake, while polymeric surface coatings that have a cationic charge can significantly enhance the cellular uptake and aggregation of nanoparticles at tumor sites [84]. Nanoparticles with a positive surface charge can take up more cancer cells and increase the drug concentration inside them [85]. Other therapeutic components, such as pigment epithelial-derived factor (PEDF), adriamycin (ADR) [86], and nucleic acid molecules [87,88], can also be prevented from aggregating and being phagocytosed by the nanoparticles, prolonging their circulation time in vivo.

### 14.2. Targeted NPs with a pH Sensitivity

As previously indicated, chemotherapy is influenced by hypoxia and an acidic TME resistance. This transmission system was created using pH-sensitive polymers that remains stable in physiological conditions and have an impact on tumor tissues by lowering pH, increasing anti-tumor effectiveness, and minimizing adverse effects. Along with pH-responsive polymers, they have basic residues or ionizable acidic residues sensitive to ionization by changing the pH of the surrounding medium. In addition to this, they are distinctively triggered by environmental pH, which results in changing the physicochemical properties such as solubility, chain conformation, surface activity, and conformational changes [89]. The pH can alter due to several physicochemical characteristics, such as polymers with imine bonds, tertiary amine bonds, amide bonds, and ionizable weak acid groups, which are acid-sensitive connecting segments. Acidic TME-responsive cancer nanotherapeutics are widely manufactured. Polyacrylic acid (PAA) is a drug carrier with carboxyl groups that hydrolyze and break in acidic TME, rupturing polymeric NPs and releasing anti-tumor drugs [90]. Lee et al. [91] utilized a cisplatin-loaded poly (acrylic acid-co-methyl methacrylate) copolymer to achieve well-targeted therapeutic benefits in the CT26 animal CRC model. Under acidic conditions, the amine bonds are easily hydrolyzed and unstable. The pH-sensitive NPs were constructed from CRC treatment based on their special pH-responsive properties. Zhang et al. [92] developed TME Doxorubicin (DOX), which was grafted onto an imine bond-based aldehyde HA and then bound to mPEG to create pH-sensitive cleavable mPEG 2k-DOX. When compared to free drugs, doxorubicin’s in vivo circulation duration is increased by around 12.5 times thanks to the PH-sensitive loading of NPs, which also efficiently targets tumor tissues to prevent toxicity. Feng et al. [93] have synthesized nanomicelles based on PEG and poly (N-(Nʹ,Nʹ-diisopropyl amino ethyl) aspartamide) (P(Asp-DIP)) and poly (lysine-cholic acid) (P(Lys-Ca)) of nano micelles. The tertiary amino group in p (Asp-DIP) is pH-responsive. In acidic TME, the released drug components are hydrophobic-hydrophilic transitions. Encapsulation with paclitaxel and superparamagnetic iron oxide in copolymers (SPIO) demonstrated that paclitaxel was delivered to tumor tissue by pH-sensitive micellar NPs in an MRI-visible drug delivery system. In acidic conditions, the boronic ester bonds [93] and hydrazone bonds [94] are unstable and commonly present in preparing pH-sensitive NPs. In an acidic environment, an increase in drug release rate is due to prepared diblock copolymers based on mPEG and polyamino acid blocks and bonds doxorubicin of deblock copolymers through pH-sensitive hydrazone bonds according to Brunato et al. [95]. The majority of alkaline chemotherapeutic medicines refuse to enter cells after protonation in acidic TME, which causes tumor cells to naturally develop resistance. In acidic hypoxic TME, the release of medications to destroy tumor cells via sensitive targeted NPs also lowers non-specific tissue harm.

### 14.3. NPs with Redox Responsiveness

In normal tissues and cells, TME redox status differs significantly in the microenvironment. The imbalance of the redox status is due to ROS and metabolic enzymes associated with glutathione being overexpressed in several subcellular structures [96]. The overall oxidative stress state is dependent on high ROS levels and high levels of glutathione (GSH), a reducing agent. Using the TME-specific redox microenvironment and GSH and ROS as stimulators of intelligent response NPs, redox-sensitive NPs can be created [92]. Cancer development, progression, and metastasis are closely associated with GSH in regulating intracellular redox homeostasis [97]. A GSH-rich TME has disulfide, the most prevalent chemical connections between NPs and anti-tumor medications, then control pharmaceuticals are reduction-responsive chemical bonds release [98]. NPs with lipoic acid (LA) and xylan (Xyl) conjugated for niclosamide (Nic) loading. Redox-responsive NPs was prepared by using thioester bonds, which help to release the loaded drug in tumor cell [99]. SN-38 coupled with redox-responsive NPs is created from ethylene glycol oligomers (OEG) via thioether linkages. Thioester bonds are responsive to both ROS and GSH, according to the results. The thioether bonds can be hydrolyzed under the oxidative influence of ROS, and they can be sulfated in the presence of GSH. In both cases, this helps in redox-responsive NPs releasing drugs to tumor sites. Under mild stimulation, conduction Di-selenium bonds are more sensitive than their counterparts, which include sulfur because they have lower bond energies. They also forcefully target drug release in response to changes in redox levels in the microenvironments, which helps to reduce non-specific tissue harm [100]. The targeted NPs, redox-responsive, access tumor cells through the ECM, which enhanced EPR effects as simple NPs. They are activated by a particular redox environment of the tumor tissue to release medications, which helps to reduce non-specific tissue harm [4]. Different nanomaterials for CRC detection and treatment have been discussed in Table 2.

## 15. Combine Nanotechnology-Based Approaches for CRC Detection and Treatment

Recent advances in nanotechnology have allowed researchers to fabricate several nanomaterials for precise diagnosis and treatment. However, the clinical applications of nanotheranostics are limited because of their complex pharmacokinetics [126]. Various nanomaterials have been verified in cancer biology and the nanocomposites include gold NPs, dendrimers, liposomes, silica NPs, and nano-emulsions, these composites have been used in imaging techniques and nano-drug delivery [127,128,129,130].

## 16. Enhancement of Imaging Techniques

Nanomaterials have been used for improving the capability of imaging techniques such as the persistent luminescence nanoparticles (PLNPs) which can be used as novel optical nanoprobes for characteristic long-lasting near-infrared (NIR) luminescence in optical imaging without autofluorescence and excitation [131]. The iron oxide-based NPs and radioisotope chelator-free NPs have been used in magnetic resonance imaging (MRI) and positron emission tomography (PET), respectively [132]. Recent studies have proved that NPs can work as integrated diagnostic and treatment agents and can be used for theranostic approaches [63].

## 17. Combined Drug Delivery

Drug delivery is the fundamental functional purpose of nanotechnology in the field of cancer. By employing NPs to create the drug delivery system, the problems of multidrug resistance, stability, efficacy, and biocompatibility have been improved. Chemotherapy and other therapies usually have serious negative outcomes, which are also minimized with the use of a formulation based on nanotechnology.

As an example, thiolated chitosan and 5-FU nanoencapsulation, a combinatorial nanomedicine agent, are non-toxic and have improved chemotherapy efficacy in CRC patients. A decrease in dose volume has also resulted from the use of nanomedicine. Compared to 5-FU nanoencapsulation, the dose used in traditional therapy was significantly greater and more hazardous [133].

Although several nanoformulations are undergoing clinical trials, the number of nanoformulations used in clinical trials against CRC is limited. Some of the nanoformulations used for the appropriate clinical trials against CRC are summarized in Table 3.

## 18. Conclusions and Future Prospective

Research on the creation of new drug delivery systems, targeted therapies, and medical devices has expanded as a result of the development of nanotechnologies. With the help of nanotechnology, medical tools have evolved from a single mode of action to multifunctional platforms, such as nano theranostics, which combines medicines and diagnostics. Nanomaterials have been successfully used in preclinical research to treat cancer.

Treatment is made simpler and quicker by combining therapy and diagnosis, or the theranostic application of nanomaterials. CRC is mostly linked to lifestyle, sex, and race, indicating that some demographics are particularly sensitive. Thus, it appears that routine CRC screening is strongly advised to stop CRC from occurring. Early detection of CRC improves survival and cancer-cure prospects. Nanomaterials’ potential for the theranostic treatment of CRC is still being explored. Numerous researchers have already documented the effective therapy of CRC in vivo in a variety of CRC model animals as well as in vitro in cancer cell lines. However, new strategies are required to enhance the existing therapies. Maybe more recent therapies employing nanomaterials to treat CRC are going to be available for clinical application. While having a lot of potentials, there are some problems with these nanotherapeutic systems’ biodistribution, localization improvement, biocompatibility, and in vivo effectiveness to treat colorectal cancer in real-time. Although the use of these nanomaterials for the treatment of colorectal cancer is still in its early stages, researchers and scientists are eager to include nanotechnological techniques in the management of CRC. The majority of targeted NP research is still in the animal testing phase at this time. From research to clinical applications, targeted NPs for CRC therapy still have a long way to go. The nanoparticles should be created to address the tumor’s biology by improving drug absorption because each patient’s tumor is unique in terms of its features and microenvironment. More in vivo preclinical investigations are needed to fully understand the mode of action of the formulations against CRC. More study is required but the development of nanoparticles for delivering drugs has significant promise for improving the standard of living and survival of CRC patients.

The significance of nanotechnology in contemporary medicine has been underlined by the thorough analysis of predicted immune toxicity assays, nanoparticle surface characterization, and quantitative comparison of encapsulated versus free drug fractions. Additionally, many nanoparticle-based studies have concentrated on the creation of methods to tailor innovative drug conjugates, diagnostics, and therapeutic devices. In addition to that, therapeutic agents, fluorescent molecules, or even magnetic materials can be programmed into nanocarriers to be released at the colorectal cancer site. This increases the bioavailability, drug solubility, stability, and tumor specificity of therapeutic agents compared to free molecular cargo.

The nanotechnology used to treat CRC has developed sufficiently in recent years to support the most current developments in tumor diagnosis and therapy, going far beyond conventional systems. With the aid of the current methodology, it can be coupled with entirely innovative therapeutic and diagnostic principles. Hence, nanomedicine will eventually be able to play a crucial part in the management of human CRC despite the numerous difficulties in the application of nanotechnology.

## Figures and Tables

**Figure 1 ijms-24-07922-f001:**
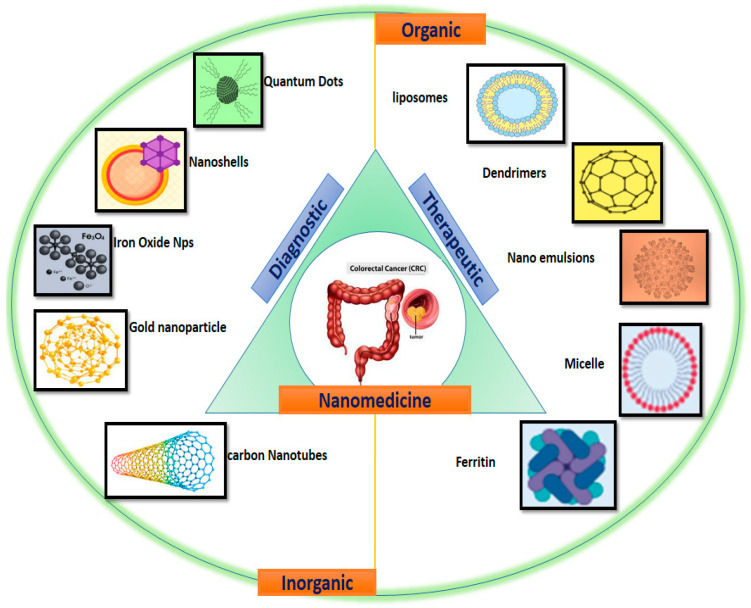
Schematic representation of different theranostics nanomedicine approaches in CRC.

**Table 1 ijms-24-07922-t001:** The development of cancer immunotherapy in CRC.

Drug/Inhibitor	Action	References
**Lentinan (LNT) and Ursolic acid (UA)**	Apoptosis was induced and immunogenic cell death was initiated in CRC	[14]
Porphyromonas gingivalis	For the treatment of cancer, M1/M2 macrophages, the growth of primary and secondary tumors in CT26 colon cancer were slowed by laser and anti-PD-1 treatment.	[15]
NLG919	NLG919-mediated suppression of indoleamine 2,3-dioxygenase 1 (IDO-1) resulted in the reversal of the immunosuppressive tumor microenvironment. The outcomes demonstrated that this method might successfully eradicate CT26 colon cancers.	[16]
IPI-549	Utilizing Ce6 as a photosensitizer in immunotherapy for colon cancer	[17]
PD-L1	Inhibit tumor growth and invasion of CRC	[18]
Cytosine-phosphate-guanine oligodeoxynucleotides (CpG ODNs)	The greatest immunological supplements	[19]
Mesoporous SiO_2_ (or Mesoporous silica, MS)	Medication delivery systems to enhance cancer treatment, very effective administration, excellent biocompatibility, simple surface modification, and self-adjuvanticity	[20]
CD166& miR-148a	A good prognostic marker for CRC	[21]

**Table 2 ijms-24-07922-t002:** Overview of CRC detection and therapy nano compositions.

Nanocompositions	Structure /Drug Loading/Encapsulation Efficiency	Application	Reference
Carbon nanotubes	Synthetic polyampholyte conjugated into single-walled carbon nanotubes (SWCNTs) for the delivery of Paclitaxel in cancerous cells	Detection and Treatment	[101]
Iron oxide nanocrystals	The diameter of iron oxide particles is with 1–100 nm. Wheat germ agglutinin (WGA) and Methotrexate SPIONS	Detection	[102,103]
CUR-CS-NP	Curcumin incorporated with chitosan nanoparticles (200–300 nm/-/80%/)	Detection and Treatment	[104]
Carbon nanotubes	Synthetic polyampholyte conjugated into single-walled carbon tubes (SWCNTs) for the delivery of paclitaxel in cancerous cells (142 nm/-/93%)	Detection and Treatment	[105]
NP SQ emcitabinel isoCA-4	Precipitates of gemcitabine, isocombretastain A-4 (isoCA-4)	Treatment	[105]
Dendrimers	Synthetic polymer with hyper-branched pattern with monomer units of regular repeats	Detection and Treatment	[106]
Gold Nanoshells	Gold surface plasmon resonant made up of silica nano core-shell and surrounded by an ultra-thin shell of gold	Detection and Treatment	[107]
Quantum dots	Semiconductor nanocrystals range from 2–10 nm in diameter	Detection and Treatment	[107]
Nanocells or PLGA nanoparticles	PLGA copolymers with different structural variants are used as efficient carriers of drug delivery	US FDA-approved therapy and detection	[108]
Liposomes	Closed and self-assembled lipid bilayer structure and colloidal (-/400–600 mg/m^2^/17%)	Detection and Treatment	[109]
Oxaliplatin/DCK and 5-FU	Nanoemulsion loaded into hydrophilic 5-FU and amphiphilic Oxaliplatin linked N-deoxycholic-L-methyl ester (DCK) (20 nm/10 mg /38.1%)	Treatment	[110]
Nanogel	In aqueous solutions Beta cyclodextrin and nanoparticles from nano gels in the presence of a cross-linker 5-FU. Nanogels are biocompatible materials and are efficient in releasing the drugs (55 nm/-/40.48%)	Treatment	[111]
PFA@PTX NPs	Poly (ferulic acid) (PFA) and PFA NPs loaded into paclitaxel (PTX) (100 nm/-/5.1–8.3%)	Treatment and Detection	[112]
SN-38 Liposome	SN 38-PA prodrug was synthesized by conjugating the SN38-C10 ester bond to the palmitic acid and encapsulated using the film dispersion method into the liposomal carrier (80.13 nm/3 mg/-)	Used for the treatment of metastatic CRC patients	[113,114]
FOLFOX	5FU, oxaliplatin incorporated into lipid nanoparticles	Treatment (efficiently treated in mice models)	[115]
5FU/PEG-PBLG	Polymeric nanoparticles loaded with 5FU (200–400 nm/20–30 g/-)	Treatment	[116]
Oxaliplatin polymeric nanoparticles	Oxaliplatin encapsulated in chitosan-coated alginate microspheres	Treatment	[117]
Chitosan-HA oxa NPs	Oxaliplatin-loaded polymeric NPs	Targeted delivery to the tumor environment	[118]
Oxaliplatin liposomes	Liposome embedding silicon microparticles	Treatment	[119]
nSN38	NCURSN38, Curcumin conjugated NPs (-/10 mg/-)	Treatment	[120]
Celecoxib conjugated NPs	Celecoxib containing Hap-Cht Nanoparticles	Treatment	[121]
Aspirin conjugated NPs	Aspirin-loaded nano exosomes (50–150 nm/5%)	Treatment	[122]
Chol-butryrate SLNP formulation	Butyric acid lipid-based nanoparticles	Treatment	[123]
Endostatin polymeric NPs	PEG-PLGA-Endostar Nanoparticles (120–150 nm/20 mg)	Detection and Treatment	[124]
NBTX3	Hafnium oxide nanoparticles (NBTXR3)	Treatment	[125]

**Table 3 ijms-24-07922-t003:** A list of FDA drugs with nanoparticles that were tested in CRC clinical studies.

S.No	Nanosystem	Drug Used	Application	FDA Approval Status	Reference
**1**	Carbon Nanoparticles	Carbon Nanoparticles	Used in CRC laparoscopic surgery	Phase I trial of 150 participants	[134]
**2**	Cyclodextrin Nanoparticles	Camptothecin	Rectal cancer, solid tumors, renal cell carcinoma, and non-small lung cell cancer	Phase I/II trial	[135]
**3**	Liposome	Vincristine	Sarcoma, colorectal cancer, neuroblastoma, acute lymphoblastic leukemia, brain tumors, and lymphoma	FDA approved	[136]
**4**	Liposome	SN38	Metastatic CRC	Phase II trial	[137]
**5**	Liposome	Aroplatin (Liposomal cisplatin analog)	Colorectal cancer	Phase I/II trial	[137]
**6**	Liposome	Doxorubicin	Colon cancer and liver metastasis	Phase II trial	[138]
**7**	Liposome	Liposome-encapsulated Irinotecan (IRI) hydrochloride PE	Second-line therapy for the metastatic CRC	Phase II trial (Subsequently terminated)	[139]
**8**	Liposome	SN 38 liposome	Metastatic CRC	Phase II trial (Subsequently terminated)	[140]
**9**	Liposome	PEGylated liposome (Narket-102), Irinotecan	Colorectal and Breast cancer	Phase III and I trail	[137]
**10**	CPX-1 liposome	Floxuridine and Irinotecan	Advanced colorectal cancer	Phase II trial 65 participants	[134]
**11**	PEP02 liposome	Leucovorin and Irinotecan and 5-FU	Metastatic CRC	Phase II trial 55 participants	[134]
**12**	MM-398	Liposomal IRI	Advanced cancer of unresectable nature	Phase Ib trial 10 participants	[134]
**13**	Nal-IRI	Irinotecan	Gastrointestinal and colorectal cancer	Phase I/II trial 64 participants	[134]
**14**	NKTR-102/IRI	Formulation of IRI conjugated with PEG/RI for prolonged release	KRAS mutant metastatic colorectal cancer	Phase II clinical trial 83 participants	[134]
**15**	Polymer	DAVANAT (Carbohydrate polymer) and 5-FU	Colorectal cancer treatment	Phase I/II trial (Subsequently terminated)	[84]
**16**	PEG-PGA polymeric micelle	SN38	Ovarian, lung, and colorectal cancers	Phase II trial	[141]
**17**	PEG-rhG-CSF	PEGylated recombinant human granulocyte colony-stimulating factor (CSF)	Solid malignant tumors (Head, lung, ovarian, colorectal, and neck cancer)	Phase IV trial 420 participants	[134]
**18**	Polymeric NPs + Cetuximab + Somatostatin analog	Somatostatin analog and combination of NPs cetuximab	Metastatic CRC	Phase I trial 30 participants	[134]
**19**	Silica NPs	Fluorescent CRGDY-PEG-Cy5.5- carbon dots	Colorectal malignancies and Breast cancer	Phase I/II trial 86 participants	[134]
**20**	Regulatory lymphocytes (Tregs); anti-CTLA-4	Ipilimumab and anti-PDL1 atezolizumab Cytotoxic antibodies expressed on the surface of Tregs	Colorectal cancer	FDA approved	[126]

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
