# Peer review of "Theranostics Nanomedicine Applications for Colorectal Cancer and Metastasis: Recent Advances"

_ijms, 2023, doi:10.3390/ijms24097922_

Round 1
Reviewer 1 Report
The novelty of the manuscript is limited, and the same studies have been reported.
English very difficult to understand/incomprehensible
Author Response
Dear reviewer,
Thanks for the note, we have modified the article and corrected it grammatically. we mainly focused on the theranostics applications and recent clinical and research findings of various types of nanomaterials in CRC detection and therapy.
Reviewer 2 Report
Accept
Author Response
Dear Reviewer,
Thank you so much for your validation.
Reviewer 3 Report
The paper entitled “Theranostics Nanomedicine Applications for Colorectal Cancer 2 and Metastasis: Recent Advances” by Babu Kasiet al. is an interesting review focusing on the recently developed strategy to treat and diagnose colorectal cancer and the related metastasis.
I have few suggestions and corrections for the Authors:
1) On lines 32-33: the Authors state “In 2020, there will be nearly 520000 new cases and 245000 deaths in Europe”. Since we currently are in 2023, it is incorrect to use the future when speaking about 2020. I suggest the Authors to modify the sentence with: “In 2020, nearly 520000 new cases and 245000 deaths in Europe were registered”.
2) On line 62: please replace “Standard methods for CRLM are curative resection,…” with “Standard therapeutic protocols for CRLM are curative resection,…”
3) On lines 76-77: the Authors state that nanomaterials are employed as drug carriers due to their passive targeting ability. However, nano-delivery systems are often appropriately functionalized with the aim of developing an actively targeted drug delivery system; I suggest the Authors to mention the possibility of modifying the nanoparticles surface for active targeting purposes.
4) Figure 1: I do not understand the image, since I think it is not clear the subdivision between “Diagnostic”, “Therapeutic” and “Nanomedicine”. These three words are not even related to the same aspect of a drug delivery system. I suggest the Authors to reconsider this figure. Moreover, the reference to this figure does not compare in the text.
5) Lines 88-89: the Authors write “IONS polymers are used as carriers for a variety of therapeutic molecules.” Few lines before, the Authors indicated that the abbreviation “IONS” stands for “Iron oxide nanoparticles”. Therefore, the sentence “Iron oxide nanoparticles polymers are used as carriers for a variety of therapeutic molecules.” is not correct, I do not understand why the Authors are speaking about polymers.
6) The abbreviations should be reported the first time they appear in the text: on line 94 the Authors explicit “magnetic resonance imaging (MRI)” but the MRI abbreviation already appeared on line 43.
7) On line 115: “small macromolecules” is a sort of oxymoron. They are small or they are macro.
8) On line 119: I suppose “Docataxel” is a typo; please replaced it with “Docetaxel”
9) The word nanoparticles is indicated with NPs or NP’s throughout the text; please uniform it.
10) On line 292: please explicit what “TME” stands for.
11) Lines 293-297 and 317-319: these sentences are very difficult to read, English style is very poor. Please reformulate this sentences in a more fluid and grammatically correct form.
12) Table 1: please add the drug loaded in the different nanosystems (it is not always reported)
13) Finally, I suggest the Authors to mention the possibility of exploiting the Boron Neutron Capture Therapy (BNCT) as a proimising therapeutic strategy for the treatment of different types of tumors, among other, colorectal cancer. This kind of therapy, besides being the object of several advanced clinical trials, open the way to the use of many different nanosystems (boron nanoparticles, liposomes, biomimetic nanoparticles, and many others). I suggest the authors to cite the following papers:
- Wittig, A.; Malago, M.; Collette, L.; Huiskamp, R.; Buhrmann, S.; Nievaart, V.; Kaiser, G.M.; Jockel, K.H.; Schmid, K.W.; Ortmann, U.; et al. Uptake of two B-10-compounds in liver metastases of colorectal adenocarcinoma for extracorporeal irradiation with boron neutron capture therapy (EORTC trial 11001). Int. J. Cancer 2008, 122, 1164–1171. https://doi.org/10.1002/ijc.23224.
- Altieri, S.; Balzi, M.; Bortolussi, S.; Bruschi, P.; Ciani, L.; Clerici, A.M.; Faraoni, P.; Ferrari, C.; Gadan, M.A.; Panza, L.; et al. Carborane Derivatives Loaded into Liposomes as Efficient Delivery Systems for Boron Neutron Capture Therapy. J. Med. Chem. 2009, 52, 7829–7835. https://doi.org/10.1021/jm900763b.
- Ailuno, G.; Balboni, A.; Caviglioli, G.; Lai, F.; Barbieri, F.; Dellacasagrande, I.; Florio, T.; Baldassari, S. Boron Vehiculating Nanosystems for Neutron Capture Therapy in Cancer Treatment. Cells 2022, 11, 4029. https://doi.org/10.3390/cells11244029
In some parts of the paper the quality of English language is very poor; in fact I suggested the Authors to revise some specific sentences, in order to make it easier for the reader to understand the paper.
Author Response
- On lines 32-33: the Authors state “In 2020, there will be nearly 520000 new cases and 245000 deaths in Europe”. Since we currently are in 2023, it is incorrect to use the future when speaking about 2020. I suggest the Authors to modify the sentence with: “In 2020, nearly 520000 new cases and 245000 deaths in Europe were registered”.
Response 1: Thanks for suggestion, sentence modified.
- On line 62: please replace “Standard methods for CRLM are curative resection,…” with “Standard therapeutic protocols for CRLM are curative resection,…”
Response 2: Thanks for suggestion, sentence replaced.
- On lines 76-77: the Authors state that nanomaterials are employed as drug carriers due to their passive targeting ability. However, nano-delivery systems are often appropriately functionalized with the aim of developing an actively targeted drug delivery system; I suggest the Authors to mention the possibility of modifying the nanoparticles surface for active targeting purposes.
Response 3: Thanks for suggestion, sentence corrected.
- Figure 1: I do not understand the image, since I think it is not clear the subdivision between “Diagnostic”, “Therapeutic” and “Nanomedicine”. These three words are not even related to the same aspect of a drug delivery system. I suggest the Authors to reconsider this figure. Moreover, the reference to this figure does not compare in the text.
Response 3: Figure 1 referred and elaborated the content regarding picture.
- Lines 88-89: the Authors write “IONS polymers are used as carriers for a variety of therapeutic molecules.” Few lines before, the Authors indicated that the abbreviation “IONS” stands for “Iron oxide nanoparticles”. Therefore, the sentence “Iron oxide nanoparticles polymers are used as carriers for a variety of therapeutic molecules.” is not correct, I do not understand why the Authors are speaking about polymers.
Response 3: The sentence was removed
- The abbreviations should be reported the first time they appear in the text: on line 94 the Authors explicit “magnetic resonance imaging (MRI)” but the MRI abbreviation already appeared on line 43.
Response 6: changed and referred at line 4
- On line 115: “small macromolecules” is a sort of oxymoron. They are small or they are macro.
Response 7: Used only macromolecule
- On line 119: I suppose “Docataxel” is a typo; please replaced it with “Docetaxel”
Response 8: replaced it with “Docetaxel”
- The word nanoparticles is indicated with NPs or NP’s throughout the text; please uniform it.
Response 9: uniformly set to NPs
- On line 292: please explicit what “TME” stands for.
Response 10: Thanks for the note, explicated “TME”
- Lines 293-297 and 317-319: these sentences are very difficult to read, English style is very poor. Please reformulate this sentences in a more fluid and grammatically correct form.
Response 11: Lines 293-297 and 317-319 modified grammatically.
- Table 1: please add the drug loaded in the different nanosystems (it is not always reported)
Response 12: Thanks for the suggestion, available dose data was added.
13) Finally, I suggest the Authors to mention the possibility of exploiting the Boron Neutron Capture Therapy (BNCT) as a proimising therapeutic strategy for the treatment of different types of tumors, among other, colorectal cancer. This kind of therapy, besides being the object of several advanced clinical trials, open the way to the use of many different nanosystems (boron nanoparticles, liposomes, biomimetic nanoparticles, and many others). I suggest the authors to cite the following papers:
- Wittig, A.; Malago, M.; Collette, L.; Huiskamp, R.; Buhrmann, S.; Nievaart, V.; Kaiser, G.M.; Jockel, K.H.; Schmid, K.W.; Ortmann, U.; et al. Uptake of two B-10-compounds in liver metastases of colorectal adenocarcinoma for extracorporeal irradiation with boron neutron capture therapy (EORTC trial 11001). Int. J. Cancer 2008, 122, 1164–1171. https://doi.org/10.1002/ijc.23224.
- Altieri, S.; Balzi, M.; Bortolussi, S.; Bruschi, P.; Ciani, L.; Clerici, A.M.; Faraoni, P.; Ferrari, C.; Gadan, M.A.; Panza, L.; et al. Carborane Derivatives Loaded into Liposomes as Efficient Delivery Systems for Boron Neutron Capture Therapy. J. Med. Chem. 2009, 52, 7829–7835. https://doi.org/10.1021/jm900763b.
- Ailuno, G.; Balboni, A.; Caviglioli, G.; Lai, F.; Barbieri, F.; Dellacasagrande, I.; Florio, T.; Baldassari, S. Boron Vehiculating Nanosystems for Neutron Capture Therapy in Cancer Treatment. Cells 2022, 11, 4029. https://doi.org/10.3390/cells11244029
Response 13: Thanks for your valuable suggestion. The Boron Neutron Capture Therapy (BNCT) as a promising therapeutic strategy for the treatment of different types of tumors, among other, colorectal cancer. It’s addressed Liposomes and dendrimers sections of the article and cited the reference.
Reviewer 4 Report
Phanindra Babu Kasi et. al. summarized the potential therapeutic strategy of nanoparticles for CRC treatment. This is a comparatively comprehensive revise, covering the major recent progress in material studies in pharmaceutics. However, there still exist some flaws in the manuscript, which I would suggest the authors to revise. Here are the comments from the reviewer:
1. In the title, the authors include metastasis as one of the aspects of the major content. However, there is no related information in the abstract.
2. Description on the materials and the therapeutic effects on CRC is too superficial. It is highly suggested that the authors should combine the principles of the materials with the correlated molecular mechanisms of CRC.
3. As the molecular mechanisms of primary cancer and metastasis are different. The authors should separately describe the materials, which are applied for primary cancer and which are used for metastasis.
4. The figure legend should provide a detailed description of the picture. Only figure caption is not enough.
5. The advantages and shortcomings of each specific material should be clearly addressed in the corresponding section.
6. As the authors intended to address the pharmacological effects of novel nanoparticles for CRC treatment, the dose information should be included. Also, the in vitro and in vivo animal studies should be addressed in detail.
7. For Table 2, the dose information should be included for all the nanoparticles, which are listed to have therapeutic potentials.
8. Which kind of mice models have been performed in the literatures? PDX or xenograft tumor cell lines? Detailed information should be included.
9. For Table 3, how many patients have been investigated? This information should be included.
10. The conclusion section should include the authors’ own way of thinking about the nanoparticles. For example, what are the limitation of current studies? What is the bottleneck for the clinical trials of these nanoparticles?
11. The limitation of this review should also be included.

Author Response
- In the title, the authors include metastasis as one of the aspects of the major content. However, there is no related information in the abstract.
Response 1: Thank you very much for the note we added the following point in the abstract “Liver and the peritoneum are where metastasis occurs most frequently. Once the tumor has metastasized to the liver, peritoneal carcinomatosis is frequently regarded as the disease's final stage. Nearly 50% of CRC patients with peritoneal carcinomatosis do not have liver metastases, nevertheless”
- Description on the materials and the therapeutic effects on CRC is too superficial. It is highly suggested that the authors should combine the principles of the materials with the correlated molecular mechanisms of CRC.
Response 2: Thanks for the suggestion, we mainly focused on the theranostics applications, recent clinical and research findings of various types of nanomaterials in CRC detection and therapy. According to the literature survey, the available information is very limited to combining the principles of the materials with the correlated molecular mechanisms of CRC
- As the molecular mechanisms of primary cancer and metastasis are different. The authors should separately describe the materials, which are applied for primary cancer and which are used for metastasis.
Response 3: Thanks for the note available data was described especially for metal nanoparticles for primary cancer and metastasis. According to an analysis of the literature, there is relatively little knowledge of particular nanomaterials for molecular mechanisms of primary cancer and metastasis.
- The figure legend should provide a detailed description of the picture. Only figure caption is not enough.
Response 4: Thanks for the suggestion, we have elaborated the content regarding the picture.
- The advantages and shortcomings of each specific material should be clearly addressed in the corresponding section.
Response 5: Thanks for the note, advantages, and a few shortcomings of each specific material described Liposomes and Dendrimers sections.
- As the authors intended to address the pharmacological effects of novel nanoparticles for CRC treatment, the dose information should be included. Also, the in vitro and in vivo animal studies should be addressed in detail.
Response 6: Thanks for the suggestion, available information on each specific material was clearly addressed.
- For Table 2, the dose information should be included for all the nanoparticles, which are listed to have therapeutic potentials.
Response 7: Thanks for the suggestion, available dose data was added.
- Which kind of mice models have been performed in the literatures? PDX or xenograft tumor cell lines? Detailed information should be included.
Response 8: Thanks for the suggestion, in the literature colorectal cancer patient-derived xenografts (PDX) model was used where LPK-PTX NPs exhibit enhanced antitumor activity and decreased systemic toxicity. The information was included in the review.
- For Table 3, how many patients have been investigated? This information should be included.
Response 9: We are glad for this note, available patient data was included.
- The conclusion section should include the author’s own way of thinking about the nanoparticles. For example, what are the limitations of current studies? What is the bottleneck for the clinical trials of these nanoparticles?
Response 10: The limitation of current studies were addressed in the conclusion section.
- The limitation of this review should also be included.
Response 11: Thanks for suggesting limitations of this review included
Round 2
Reviewer 1 Report
Most of the questions have been addressed. However, the progress of cancer immunotherapy need to be systematically summarized. These reports can be referred: doi: 10.3389/fimmu.2022.1024931; doi: 10.1177/17246008211005473; doi: 10.3389/fimmu.2021.653836; doi: 10.3389/fonc.2021.822745; doi: 10.2174/0929867326666191004164041; https://doi.org/10.1186/s12863-019-0795-z. The resolution of the figures should be improved, and the format of the reference list needs to be checked again.
Author Response
Dear reviewer,
Thank you very much for your insightful remarks and recommendations. We have systematically summarized the progress of cancer immunotherapy in a table format in the manuscript. We've improved the figure resolution, and the format of the reference list was thoroughly checked and arranged in an orderly format.
Reviewer 4 Report
The manuscript has been greatly improved with the author's careful revision. I would recommend accepting this manuscript at the present form.
Author Response
Dear Reviewer,
Thank you very much for your insightful remarks and recommendations.